# The Association between the Severity of Dysmenorrhea and Psychological Distress of Women Working in Central Tokyo—A Preliminary Study

**DOI:** 10.3390/ijerph20217021

**Published:** 2023-11-05

**Authors:** Kumi Matsumura, Kanami Tsuno, Masumi Okamoto, Akiko Takahashi, Akio Kurokawa, Yuko Watanabe, Honami Yoshida

**Affiliations:** 1Kanagawa Prefectural Government, Yokohama 231-0021, Kanagawa, Japan; arakawa.3nj6@pref.kanagawa.lg.jp; 2School of Health Innovation, Kanagawa University of Human Services, Research Gate Building TONOMACHI 2-3D, 3-25-10 Tonomachi, Kawasaki 210-0821, Kanagawa, Japan; k.tsuno-wm4@kuhs.ac.jp (K.T.); a.kurokawa-4k5@kuhs.ac.jp (A.K.); 3Center for Innovation Policy, Kanagawa University of Human Services, Kawasaki 210-0821, Kanagawa, Japan; m.okamoto-mw7@kuhs.ac.jp (M.O.); y.watanabe-g8m@kuhs.ac.jp (Y.W.)

**Keywords:** dysmenorrhea, psychological distress, mental health, working women

## Abstract

This study aims to clarify the association between the severity of dysmenorrhea and psychological distress among working women in central Tokyo and examine the effect modification of job stressors. The participants in this cross-sectional study were 312 women who had undergone health check-ups in the “Marunouchi Hokenshitsu” project. The severity of dysmenorrhea was defined as the degree of daily life disturbance with menstrual pain, and the outcome variable was the K6 scores. To assess the association of psychological distress with the severity of dysmenorrhea, multiple regression analyses were performed. The results revealed that 18.3% of the 289 working women were in the moderate/severe group of dysmenorrhea. In multiple regression analysis, moderate/severe dysmenorrhea was significantly associated with higher levels of psychological distress, but the significance disappeared after adjusting for gynecology such as premenstrual syndrome (PMS) and workplace-related factors. The degree of job control was significantly associated with lower levels of psychological distress and may modify psychological distress caused by dysmenorrhea. Moderate/severe dysmenorrhea may be associated with higher levels of psychological distress in working women, and psychological symptoms of PMS) and the degree of job control were possible effect factors, and there may be effect modification by the degree of job control.

## 1. Introduction

Dysmenorrhea, or menstrual pain, is defined as pain during the menstrual cycle and is the most common menstrual symptom among adolescent girls and young women. Most adolescents experiencing dysmenorrhea have primary dysmenorrhea, defined as painful menstruation in the absence of pelvic pathology [1]. Dysmenorrhea is treated as synonymous with menstrual pain in some literature, with prevalence rates varying from 15 to 95% and menstrual pain ranging from mild to limiting daily life [2]. This wide variation in the prevalence of dysmenorrhea is because of several scales to assess the syndrome. In some literature, the intensity of menstrual pain is defined using the Visual Analog Scale [3,4], whereas in others, the severity is defined by the impact of daily life and use of analgesics [5]. On the other hand, it has been shown that the severity of menstrual pain may not be related to the impact of daily life or the need for analgesics [6], and prevalence rates vary depending on how the severity of dysmenorrhea is assessed.

Dysmenorrhea affects mental health as well as physical health. It was found that anxiety and depression levels were higher in adolescents with dysmenorrhea [7]. In university students, primary dysmenorrhea without gynecological disease is associated with a high level of psychological distress [8], and secondary dysmenorrhea due to gynecological disorders is associated with a high level of psychological distress [9]. However, studies focusing on dysmenorrhea in working women are limited. In working women, workplace-related factors such as working hours associated with menstrual symptoms and work productivity [10,11], and job stressors such as low job control, less peer support, and job insecurity for employment are associated with the severity of dysmenorrhea [12,13]. However, previous studies have not estimated the effect of modification of job stress on the association between the severity of dysmenorrhea and psychological distress in working women. Identifying effect modifiers of the association between the severity of dysmenorrhea and psychological distress may promote coping behaviors and workplace considerations for dysmenorrhea, which could be utilized to alleviate psychological distress among working women.

The objective of this study was to clarify the association between the severity of dysmenorrhea and psychological distress among working women in central Tokyo and to examine the associated factors and the effect modification of job stress. We focused on working women in central Tokyo, where approximately 3.75 million women are employed [14], and the average ages at first marriage (30.4 years) and first childbirth (32.2 years) are the highest in Japan [15]. As the later age of first childbirth increases, the number of menstrual periods before first childbirth increases the risk of developing serious gynecological disorders [16].

## 2. Materials and Methods

### 2.1. Procedures and Participants

The participants in this cross-sectional study were 312 women in their 20s to 60s working in the Marunouchi area (Business districts in Chiyoda Ward, Tokyo) who underwent the original health check-up by Clairage Tokyo Ladies’ Health Check-up Clinic.

The study included 289 working women, after excluding those not in employment (*n* = 3); absent from work in the past month (*n* = 3); medicated for mental illness (*n* = 8); pregnant or possibly pregnant (*n* = 3); or menopausal (*n* = 7) (1 participant had overlapping exclusion criteria) based on their answer to the survey. We included women in their 60s if they were not in menopause yet.

The check-up project was operated by Femmes Médicaux, Co. (Tokyo, Japan), in the “Marunouchi Hokenshitsu” project run by the Mitsubishi Estate Company, Limited (Tokyo, Japan), which is responsible for the development of the Marunouchi area. The health check-up project was conducted between 2 and 16 October 2021. Mitsubishi Estate Company, Limited, invited nine companies in the Marunouchi area to participate individually and publicized the event through press releases and a website. Participants via companies participated for free, and those not via companies participated for a fee. Participants who applied for this health check-up project and consented to participate in this study were given medical interviews, body composition measurements, blood tests, transvaginal ultrasound examinations, gynecological examinations, etc. We excluded those not in employment, absent from work in the past month, medicated for mental illness, pregnant or possibly pregnant, or menopausal. The medical interview was held at the clinic on weekdays and the interview was conducted in a private room with medical staff as the interviewers. All data in this study were collected during the health check-up project. These anonymized data were received by the School of Health Innovation, Kanagawa University of Human Services, and subjected to statistical analysis.

### 2.2. Measurements

#### 2.2.1. Dysmenorrhea

In this study, the presence of gynecological disorders was assessed by transvaginal ultrasonography. However, as primary dysmenorrhea without a gynecological disorder cannot be medically diagnosed, the severity of dysmenorrhea was defined based on previous studies as the degree of daily life disturbance with menstrual pain during menstruation [5,6]. First, participants who responded with answers other than “no symptoms” regarding the frequency of menstrual pain (“rarely”, “sometimes”, “always”) were further asked about the intensity of symptoms (“do not bother”, “can be tolerated”, “difficulties in daily life/work”). Participants were divided into two groups: a none/mild group, as they reported “no symptoms”, “do not bother”, or “can be tolerated”; and a moderate/severe group, as they reported “difficulties in daily life”.

#### 2.2.2. Psychological Distress

Psychological distress was measured by the K6 scale, a widely used worldwide screening measure of psychological distress in diverse populations. The Japanese version of the K6 scale has been validated [17]. Participants indicated how often they had six different feelings or experiences during the past 30 days: “nervous”, “hopeless”, “restless or fidgety”, “so depressed that nothing could cheer you up”, “that everything was an effort”, and “worthless”. Items responses were measured on a five-point Likert scale: 4 (all of the time), 3 (most of the time), 2 (some of the time), 1 (a little of the time), and 0 (none of the time). The minimum total score for the options is 0, and the maximum total score is 24, with higher scores indicating higher levels of psychological distress. Although a cut-off of 13 points or more is considered valid for screening severe mental illness [18], a study in Japan suggested that a K6 total score of 5 or more is the optimal cut-off point for psychological distress [19]. The participants in this study with a cut-off point of 5 or more points were defined as having psychological distress.

#### 2.2.3. Job Stressors

Job stressors were measured using some items from the New Brief Job Stress Questionnaire (BJSQ), which is also used for employee stress checks and has been tested for reliability and validity in Japan [20,21], measuring quantitative job overload (3 items), qualitative job overload (3 items), and job control (3 items). Examples of each item are as follows: quantitative job overload is “I have a tremendous amount of work to do”; qualitative job overload is “I have to pay very careful attention”; and job control is “I can work at my own pace”. All items of the BJSQ are available on the website of the Ministry of Health, Labour, and Welfare [22]. Items responses are measured on a four-point Likert-type scale: 4 (not at all), 3 (somewhat), 2 (moderately so), and 1 (very much so). Reversal items for job stress were scored by replacing 1–4 points with 4–1 points, and the average score was used.

#### 2.2.4. Covariates

All data in this study were collected during the health check-up project by online survey questionnaire.

Covariates were selected based on previous studies [6,9,12,23,24,25]: age; body mass index (BMI); educational status (university graduate, others); occupation (clerical job, service job, professional job, management job, others); marital status (married, unmarried); history of pregnancy (yes, no) and the number of children given birth to (0, 1, 2, 3); and gynecology and workplace-related factors, including working hours and sleeping hours during the workday.

The following information was selected as gynecology-related factors: menstrual cycle (≤24, 25–38, ≤39 days, irregularity, no answer); menstrual flow (heavy, normal, low, no answer); experience of seeing obstetricians and gynecologists (yes, no); taking oral contraceptive pill (yes, no); current or history of gynecological disorders (yes, no); knowledge of dysmenorrhea (participants were divided into two groups: no knowledge of symptoms group if they reported “do not know” and “heard of the name,” and a knowledge of symptoms group if they reported “know the symptoms but not how to deal with it” and “know more about the symptoms and how to deal with it”); physical symptoms of premenstrual syndrome (PMS); and psychological symptoms of PMS, anemia and blood iron status. Physical and psychological symptoms of PMS were divided into two groups according to severity, similar to the method used for dysmenorrhea (none/mild, moderate/severe). Anemia was classified according to hemoglobin, total iron binding capacity, and serum ferritin by blood tests (normal, anemia with iron deficiency, anemia without iron deficiency, and iron deficiency without anemia) based on the guidelines for the treatment of anemia through the appropriate use of iron supplements [26].

The following information was selected as workplace-related factors: working hours (total of fixed working hours and overtime hours per day) and sleeping hours (actual sleeping hours per working day in the past month).

### 2.3. Statistical Analyses

Group differences by severity of dysmenorrhea were assessed using the chi-square test for categorical variables and the Wilcoxon rank sum test for continuous variables to check whether they were normally distributed. To assess the association of psychological distress with the severity of dysmenorrhea, multiple regression analysis was performed, and standard partial regression coefficients (β) and standard errors (SE) were calculated. In the multiple regression analysis, the Variance Inflation Factor (VIF) of all variables was checked to ensure that they were less than 10 to account for multicollinearity issues. Model 1 assessed the association between dysmenorrhea severity or each covariate and psychological distress in a single regression analysis. Model 2 was adjusted for the following basic attributes: age, BMI, and marital status. Model 3 was adjusted for the following gynecology-related factors: experience of seeing obstetricians and gynecologists; taking oral contraceptive pills; gynecological disorder; knowledge of dysmenorrhea; and physical and psychological symptoms of PMS, in addition to the covariates in Model 2. Model 4 was adjusted for the following work-related factors: working hours; sleeping hours; quantitative job overload; qualitative job overload; and job control, in addition to the covariates in Model 3. In Model 5, an interaction term between dysmenorrhea severity and job control was created and adjusted to assess the interaction between dysmenorrhea severity and workplace-related factors. STATA SE (Standard Edition version 17.0) was used for all statistical analyses, and *p* < 0.05 was considered statistically significant. The post hoc power of the study was estimated using G*Power software (version 3.1) with an α-error of 0.05.

## 3. Results

### 3.1. Characteristics of the Study Participants

The study included 289 working women, after excluding those not in employment (*n* = 3); absent from work in the past month (*n* = 3); medicated for mental illness (*n* = 8); pregnant or possibly pregnant (*n* = 3); or menopausal (*n* = 7) (1 participant had overlapping exclusion criteria). The characteristics of the study participants and their severity of dysmenorrhea are summarized in Table 1. Table 1 shows that 81.7% of the study participants were in the none/mild group, and 18.3% were in the moderate/severe group of dysmenorrhea. The most common age group of the entire study participants was in their 20s (33.2%). The moderate/severe group had significantly higher rates of heavy menstrual flow (17.0%), moderate/severe physical symptoms of PMS (58.5%), moderate/severe psychological symptoms of PMS (37.7%), experience of seeing obstetricians and gynecologists (79.3%), knowledge of symptoms (66.0%), and psychological distress (67.9%) than the none/mild group. In addition, job stress in the moderate/severe group was significantly lower than in the none/mild group (median, 25th percentile, 75th percentile: 3, 2.7, and 3).

### 3.2. Severity of Dysmenorrhea and Psychological Distress

Table 2 shows the results of a multiple regression analysis assessing the association between dysmenorrhea and psychological distress. In the analysis, variables with a VIF of 3 or more were excluded from the analysis to account for multicollinearity issues. In a single regression analysis (Model 1), moderate/severe dysmenorrhea (β = 3.168, *p* < 0.001), moderate/severe physical symptoms of PMS (β = 3.234, *p* < 0.001), moderate/severe psychological symptoms of PMS (β = 4.985, *p* < 0.001), and taking oral contraceptive pill (β = 1.672, *p* < 0.05) were significantly associated with a higher level of psychological distress. Longer sleeping hours (β = −0.583, *p* < 0.05) and higher levels of job control (β = −2.504, *p* < 0.001) were also significantly associated with a lower level of psychological distress.

After adjusting for basic attributes (Model 2), only moderate/severe dysmenorrhea was significantly associated with a higher level of psychological distress (β = 3.151, *p* < 0.001). When further adjusted for gynecology-related factors (Model 3), the association between moderate/severe dysmenorrhea and a higher level of psychological distress was marginally significant (β = 1.548, *p* < 0.10), and moderate/severe psychological symptoms of PMS were significantly associated with a higher level of psychological distress (β = 3.978, *p* < 0.001). When adjusted for workplace-related factors (Model 4), the association between moderate/severe dysmenorrhea and psychological distress was no longer significant, and moderate/severe psychological symptoms of PMS (β = 3.978, *p* < 0.001) and a gynecological disorder (β = 1.521, *p* < 0.05) were significantly associated with a higher level of psychological distress. Moreover, higher levels of job control were significantly associated with a lower level of psychological distress (β = −1.900, *p* < 0.001). When the interaction term between moderate/severe dysmenorrhea and job control was adjusted for Model 5, the factors significantly associated with Model 4 remained significant and there was no significant association between the interaction term and psychological distress. Statistical power was 0.963 for Model 2, 0.999 for Model 3, 0.999 for Model 4, and 0.999 for Model 5.

## 4. Discussion

This study examined factors associated with the severity of dysmenorrhea and psychological distress, as well as the effect modification of job stressors among working women in central Tokyo. In this study, 18.3% of the study participants had moderate/severe dysmenorrhea, and this group had significantly higher rates of psychological distress (67.9%) than the none/mild group. Moderate/severe dysmenorrhea was significantly associated with higher levels of psychological distress, but the significance disappeared after adjusting for gynecology-related and workplace-related factors in multiple regression analyses. Of the adjusted factors, moderate/severe psychological symptoms of PMS and the degree of job control were significantly associated with the degree of psychological distress in all models. The results suggested that moderate/severe psychological symptoms of PMS and the degree of job control were possible effect factors and there might be effect modification by the degree of job control.

Regarding the association between dysmenorrhea and psychological distress, in multiple regression analysis, moderate/severe dysmenorrhea was significantly associated with higher psychological distress in a single regression analysis (Model 1), which remained significant after adjustment for basic attributes (Model 2). However, the association was no longer significant after adjusting for confounding factors with the basic attributes and gynecology-related factors (Model 3) and further adjustment for workplace-related factors (Models 4 and 5). This suggests that dysmenorrhea was associated with high levels of psychological distress among working women but could be modified by other factors. Of the adjusted factors, moderate/severe psychological symptoms of PMS were associated with higher levels of psychological distress in all models. Previous research has shown that the prevalence of PMS increased according to the severity of dysmenorrhea [24] and it was possible that the psychological symptoms of PMS may be a confounding factor in the association between dysmenorrhea and psychological distress. In addition, gynecological disorders were significantly associated with higher psychological distress when adjusting for workplace-related factors (Models 4 and 5), suggesting that gynecological disorders may interact with workplace-related factors to increase psychological distress. In order to mitigate psychological distress among working women, it is necessary to modify not only dysmenorrhea but also PMS and other symptoms associated with menstruation, and establish workplace considerations in the case of gynecological disorders.

The moderate/severe dysmenorrhea group tended to have lower levels of job control and higher levels of job control were significantly associated with lower levels of psychological distress in all models. It suggested that the degree of job control may modify psychological distress due to the severity of dysmenorrhea, but the interaction between the severity of dysmenorrhea and the degree of job control was not significant. A previous study has also shown that the degree of job control is associated with menstrual pain [12], indicating that increasing the degree of job control could make it easier for working women to adopt coping behavior towards dysmenorrhea and mitigate psychological distress due to the severity of dysmenorrhea.

Moderate/severe dysmenorrhea was prominent in the late 20s and 30s. The group had significantly higher rates of experience of seeing obstetricians and gynecologists (79.3%) and knowledge of symptoms (66.0%) than the none/mild group. Although dysmenorrhea is more common in young women in their teens and twenties and most are considered to have primary dysmenorrhea without gynecological disorders [1], it can lead to the development and severity of gynecological disorders if left untreated. The largest number of subjects in this study were in their late 20s and 30s. The results are unique in that they focus on women in central Tokyo who have a later childbearing age. Single regression analysis also showed that taking oral contraceptive pills was associated with higher levels of psychological distress. These results suggest that many people may adopt coping behaviors if they feel strongly that the symptoms interfere with their daily lives or cause psychological distress. In addition, working women in central Tokyo in their late 20s and 30s tend to be older at first marriage and first childbirth. It is crucial to facilitate access to treatment and coping behaviors at the appropriate time, depending on the severity of symptoms, rather than adopting coping behaviors when symptoms become more intense. Previous research has shown that women find discussing menstrual pain with their colleagues difficult and need workplace considerations and promotion of coping behaviors according to the severity of their symptoms and wishes [27]. Social support and flexible working arrangements, such as identifying women’s symptoms and encouraging them to take time off to facilitate coping behavior, as well as promoting leave when needed and access to treatments, are important. For example, working women could be asked to complete a menstrual symptoms check together with a stress check, and if they have symptoms, they could be encouraged to inform their managers, and managers could be required to include information on menstrual symptoms in the training they are required to attend.

This study has several limitations. First, participants were not randomly selected, and active collaborator bias may have existed since participants were relatively health-conscious. Also, participants via companies participated for free, and those not via companies participated for a fee, which may have caused selection bias. For participants with a higher prevalence of pain, and who wish to have healthcare, their willingness to pay to participate may lead to more detail in their input in the questionnaire. Thus, the results cannot be representative of 3.75 million working women in Tokyo [14], although the statistical power of the multiple regression analysis in this study is above 0.85 for all models, which seems to be substantially sufficient. Second, this study was cross-sectional and could not identify a causal association between psychological distress and dysmenorrhea. Future research designs on dysmenorrhea, job control, and psychological symptoms of PMS should be designed to identify the causal association to analyze the interactions between these factors. Third, using a single questionnaire rather than a validated one limited the accuracy of the measurement of dysmenorrhea and job stress. The scale for assessing the severity of dysmenorrhea was not clearly defined, and further research is needed to develop a more sensitive scale and investigate the association between dysmenorrhea and job stress. Fourth, social support is a protective factor for working women with higher psychological distress [28,29], but data on social support were not analyzed in this study. This study showed that increasing the degree of job control may facilitate coping behaviors for dysmenorrhea in working women and alleviate psychological distress due to the severity of dysmenorrhea. Further research is needed on social support to promote coping behaviors and access to treatments that may modify the association between dysmenorrhea and psychological distress. In addition, although an objective assessment of psychological distress was carried out in this study, in order to link this to specific considerations in the workplace, it is necessary to focus on subjective assessments to determine their concerns and feelings and to conduct other studies that propose individualized and specific support through interviews.

## 5. Conclusions

Moderate/severe dysmenorrhea may be associated with higher levels of psychological distress in working women, but in the association between dysmenorrhea and psychological distress, psychological symptoms of PMS and the degree of job control were possible effect factors, and there may be effect modification by the degree of job control.

## Figures and Tables

**Table 1 ijerph-20-07021-t001:** Participant characteristics of the severity of dysmenorrhea.

		All (*n* = 281)		None/Mild (*n* = 236, 81.7%)		Moderate/Severe (*n* = 53, 18.3%)		*p* Value *
		*n*	%	*n*	%	*n*	%	
Age (years)	Mean(SD)		34.8 (8.0)		35.5 (8.1)		31.9 (6.8)	0.08
	20–24	18	6.2	11	4.7	7	13.2	
	25–29	78	27.0	60	25.4	18	34.0	
	30–34	63	21.8	52	22.0	11	20.8	
	35–39	43	14.9	34	14.4	9	17.0	
	40–44	49	17.0	44	18.6	5	9.4	
	24–49	26	9.0	24	10.2	2	3.8	
	50–61	12	4.2	11	4.7	1	1.9	
BMI	<18.5	45	15.7	38	16.1	7	13.2	0.72
	18.5–24.9	223	77.2	182	77.1	41	77.4	
	≥25	21	7.3	16	6.8	5	9.4	
Academic background	University graduate	245	84.8	196	83.1	49	92.5	0.09
	Other	44	15.2	40	17.0	4	7.6	
History of pregnancy	Yes	79	27.3	66	28.0	13	24.5	0.61
	No	210	72.7	170	72.0	40	75.5	
Marital status	Married	105	36.3	90	38.1	15	28.3	0.18
	Unmarried	184	63.7	146	61.9	38	71.7	
Number of children given birth	0	225	77.9	181	76.7	44	83.0	0.147
	1	34	11.8	26	11.0	8	15.1	
	2	27	9.3	26	11.0	1	1.9	
	3	3	1.0	3	1.3	0	0.0	
Menstrual cycle(days)	24≥	10	3.5	10	4.2	0	0.0	0.24
	25–38	224	77.5	177	75.0	47	88.7	
	≥39–	2	0.7	2	0.9	0	0.0	
	Irregularity	33	11.4	29	12.3	4	7.6	
	No answer	20	6.9	18	7.6	2	3.8	
Menstrual flow	Heavy	31	10.7	22	9.3	9	17.0	<0.05
	Normal	207	71.6	166	70.3	41	77.4	
	Low	40	13.8	38	16.1	2	3.8	
	No answer	11	3.8	10	4.2	1	1.9	
Physical symptoms of PMS	Moderate/severe	51	17.7	20	8.5	31	58.5	<0.001
	None/mild	238	82.4	216	91.5	22	41.5	
Psychological symptoms of PMS	Moderate/severe	41	14.2	21	8.9	20	37.7	<0.001
	None/mild	248	85.8	215	91.1	33	62.3	
Taking oral contraceptive pill	Yes	42	14.5	33	14.0	9	17.0	0.58
	No	247	85.5	203	86.0	44	83.0	
Gynecological disorder	Yes	84	29.1	69	29.2	15	28.3	0.89
	No	205	70.9	167	70.8	38	71.7	
Anemia and blood iron status	Normal	216	74.7	176	74.6	40	75.5	0.57
	Anemia with iron deficiency	41	14.2	32	13.6	9	17.0	
	Anemia without iron deficiency	7	2.4	7	3.0	0	0.0	
	Iron deficiency without anemia	25	8.7	21	8.9	4	7.6	
Experience of seeing obstetricians and gynecologists	Yes	173	59.9	131	55.5	42	79.3	<0.05
	No	116	40.1	105	44.5	11	20.8	
Knowledge of dysmenorrhea	Knowledge of symptoms	144	49.8	109	46.2	35	66.0	<0.05
	No knowledge of symptoms	145	50.2	127	53.8	18	34.0	
Occupation	Clerical job	177	61.3	147	62.3	30	56.6	0.63
	Service job	14	4.8	10	4.2	4	7.6	
	Professional job	57	19.7	46	19.5	11	20.8	
	Management job	39	13.5	32	13.6	7	13.2	
	Others	2	0.7	1	0.4	1	0.4	
Working hours	−7	8	2.8	6	2.5	2	3.8	0.21
	7–8	21	7.3	18	7.6	3	5.7	
	8–9	70	24.2	61	25.9	9	17.0	
	9–10	88	30.5	65	27.5	23	43.4	
	10–	102	35.3	86	36.4	16	30.2	
Sleeping hours	−5	12	4.2	10	4.2	2	3.8	0.92
	5–6	48	16.6	38	16.1	10	18.9	
	6–7	107	37.0	87	36.9	20	37.7	
	7–8	95	32.9	80	33.9	15	28.3	
	8–	27	9.3	21	8.9	6	11.3	
Job stressors								
Quantitative job overload	Median (25th percentile, 75th percentile)	3 (2.7, 3.3)		3 (2.3, 3.3)		3 (2.7,3.7)		0.22
Qualitative job overload		2.7 (2.3, 3.3)		2.7(2.3, 3.3)		3 (2.7, 3.3)		0.08
Job control		3 (2.7, 3.3)		3 (2.7, 3.3)		3 (2.7, 3)		<0.05
Psychological distress (K6 score)	0–4	151	52.3	134	56.8	17	32.1	<0.05
	5–24	138	47.8	102	43.2	36	67.9	

* Calculated among dysmenorrhea severity groups using the chi-square test and the Wilcoxon rank-sum tests.

**Table 2 ijerph-20-07021-t002:** Multiple regression analysis for psychological distress.

Variables	Value	Model 1 (Unadjusted)			Model 2			Model 3			Model 4			Model 5		
		β	SE	*p* Value	β	SE	*p* Value	β	SE	*p* Value	β	SE	*p* Value	β	SE	*p* Value
Dysmenorrhea	0 = None/mild, 1 = Moderate/severe	3.168	0.733	<0.001	3.151	0.748	<0.001	1.548	0.841	0.07	1.254	0.860	0.15	1.014	0.87	0.25
Age(years)	Primary Value	−0.012	0.037	0.748	0.027	0.038	0.49	0.046	0.039	0.24	0.035	0.044	0.42	0.04	0.043	0.36
BMI	Primary Value	0.096	0.108	0.379	0.049	0.108	0.65	−0.038	0.105	0.72	−0.024	0.110	0.83	−0.052	0.111	0.64
Marital status	0 = Married, 1 = Unmariied	−1.009	0.605	0.097	−0.920	0.615	0.14	−0.72	0.592	0.23	−0.297	0.649	0.65	−0.169	0.651	0.80
Physical symptoms of PMS	0 = None/mild, 1 = Moderate/severe	3.234	0.743	<0.001				1.125	0.851	0.19	1.002	0.872	0.25	0.998	0.869	0.25
Psychological symptoms of PMS	0 = None/mild, 1 = Moderate/severe	4.985	0.785	<0.001				3.978	0.840	<0.001	4.535	0.903	<0.001	4.611	0.901	<0.001
Experience of seeing obstetricians and gynecologists	0 = No, 1 = Yes	1.114	0.593	0.06				0.26	0.581	0.66	0.269	0.623	0.67	0.281	0.621	0.65
Knowledge of dysmenorrhea	0 = None/mild, 1 = Moderate/severe	0.522	0.584	0.37				−0.373	0.567	0.51	−0.402	0.600	0.50	−0.449	0.599	0.45
Taking oral contraceptive pill	0 = No, 1 = Yes	1.672	0.824	<0.05				1.378	0.834	0.10	1.517	0.872	0.08	1.622	0.871	0.06
Gynecological disorder	0 = No, 1 = Yes	1.176	0.641	0.07				1.011	0.627	0.11	1.521	0.681	<0.05	1.542	0.679	<0.05
Working hours	continuous	0.018	0.131	0.89							−0.005	0.134	0.97	0.004	0.133	0.98
Sleeping hours	Primary Value	−0.583	0.288	<0.05							−0.203	0.323	0.53	−0.184	0.322	0.57
Job stressors																
Quantitative job overload	Primary Value	0.666	0.411	0.11										0.199	0.579	0.73
Qualitative job overload	Primary Value	0.994	0.507	0.05										0.517	0.617	0.40
Job control	Primary Value	−2.504	0.491	<0.001										−1.939	0.496	<0.001
Interaction (dysmenorrhea × job control)														−2.119	1.32	0.11
	R^2^					0.070			0.177			0.290			0.298	
	adjusted R^2^					0.057			0.147			0.243			0.249	
	*p* value					<0.001			<0.001			<0.001			<0.001	

## Data Availability

Data for this study cannot be publicly shared due to privacy and ethical constraints. We are committed to transparency and have provided a detailed methodology in the article to support reproducibility. For specific inquiries, please contact correspondent author.

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
