# Peer review of "The Association between the Severity of Dysmenorrhea and Psychological Distress of Women Working in Central Tokyo—A Preliminary Study"

_ijerph, 2023, doi:10.3390/ijerph20217021_

Round 1

Reviewer 1 Report

Comments and Suggestions for Authors

A well written manuscript 

Just some minor correction may be need in the methodology section: 

Line 81: Participants via companies participated for free, and those not via companies participated for a fee. -Could it be that the participant who paid may have been more detailed in their input, may have a higher prevalence of pain, and wishes to be 'investigated' and therefore opted in and willing to pay to participant? The authors need to put this into consideration. 

Where was the medical interview held? At the participants’ home, or office? Weekends weekday? During their leisure time? Was the interview done in a private room? Who were the interviewers? 

What will be offer (in terms of medical support) for women who did express severe dysmenorrhoea? 

Line 93: Dysmenorrhoea Which and what was the definition of dysmenorrhea in this study? 

Author Response

Thank you very much for taking the time to review this manuscript. I am grateful for your contribution.  Please find the detailed responses below and attached file, the corresponding revision in the re-submitted files.

3. Point-by-point response to Comments and Suggestions for Authors

Comments 1: Line 81: Participants via companies participated for free, and those not via companies participated for a fee. -Could it be that the participant who paid may have been more detailed in their input, may have a higher prevalence of pain, and wishes to be 'investigated' and therefore opted in and willing to pay to participate? The authors need to take this into consideration.

Response 1: Thank you for pointing this out. We agree with this comment. Therefore, we have made a change to explain the possible bias in the limitation section. Please check the revised manuscript and this change can be found in line 294.

“Also, participants via companies participated for free, and those not via companies participated for a fee, which may cause selection bias. For participants with a higher prevalence of pain, and who wish to have healthcare, their willingness to pay to participate may lead to more detail in their input in the questionnaire.”

Comments 2:

Where was the medical interview held? At the participants’ home, or office? Weekends weekday? During their leisure time? Was the interview done in a private room? Who were the interviewers?

Response 2: Agree. I have changed methods to emphasize this point. The changes made in the revised manuscript are in line 85.

“The medical interview was held at the clinic on weekdays and the interview was done in a private room with medical staff as the interviewers.”

Comments 3:

What will be offer (in terms of medical support) for women who did express severe dysmenorrhoea?

Response 3: Thank you for the comments. Your question is to the point. In Japan, since medical checkups are self-funded and cannot be covered by insurance and treated on the same day, we do not provide treatment during the checkups.  Therefore, we did not include information about what treatments are available for dysmenorrhea in this paper. Generally, we ask participants with dysmenorrhea who have strong symptoms to see us again on a different day and recommend some treatments, for example, treatment with low-dose pills, Chinese herbal medicine, or counseling.

Comments 4:

Line 93: Dysmenorrhoea Which and what was the definition of dysmenorrhea in this study?

Response 4: Thank you for bringing up such a basic clarification. I have changed the introduction section to emphasize this point. The changes made in the revised manuscript are in line 29.

“is defined as pain during the menstrual cycle and”

Reviewer 2 Report

Comments and Suggestions for Authors

This paper investigates the association between the severity of dysmenorrhea and psychological distress of working women. The authors found that level of dysmenorrhea was significantly associated with higher levels of psychological distress. This is an important topic, because many women experience dysmenorrhea.

Introducion:

The most important information about the studied phenomenon is provided. The purpose of the research was outlined and the possible causes were explained in the introduction.

Material and methods:

The study group is very small, which may affect the statistical results. I have some doubts about the size of the group and the age of the respondents. Younger people at the beginning of their career may experience different types of stress than older people. In my opinion, the study group should be of a similar age. It's worth considering adding "preliminary study" to the title.

Results and discussion:

The results were presented correctly. The authors carefully presented the limitations of the study. Nevertheless, the limitations of research are significant.

Line 297. A period is missing after the sentence "which seems to be substantially sufficient".

References:

Please check the formatting according to the journal requirements. Sometimes the names of journals are written as abbreviations, and sometimes as full names.

Author Response

Thank you very much for taking the time to review this manuscript. I am grateful for your contribution.  Please find the detailed responses below and the corresponding revision in the re-submitted file.

3. Point-by-point response to Comments and Suggestions for Authors

Comments 1: This paper investigates the association between the severity of dysmenorrhea and psychological distress of working women. The authors found that level of dysmenorrhea was significantly associated with higher levels of psychological distress. This is an important topic, because many women experience dysmenorrhea.

Response 1: Thank you so much for understanding the significance of this study. It was very encouraging.

Comments 2: Introducion:

The most important information about the studied phenomenon is provided. The purpose of the research was outlined and the possible causes were explained in the introduction.

Response 2: Thank you so much for your positive feedback on this study. We were so relieved to know it is understandable.

Comments 3: Material and methods:

The study group is very small, which may affect the statistical results. I have some doubts about the size of the group and the age of the respondents. Younger people at the beginning of their career may experience different types of stress than older people. In my opinion, the study group should be of a similar age. It's worth considering adding "preliminary study" to the title.

Response 3: Thank you for pointing this out. I agree with this comment. Therefore, I have added “a preliminary study” to the title.

Comments 4: Results and discussion:

The results were presented correctly. The authors carefully presented the limitations of the study. Nevertheless, the limitations of research are significant.

Line 297. A period is missing after the sentence "which seems to be substantially sufficient".

Response 4: Thank you for pointing this out. We agree with this comment. Therefore, we have added the period in line 297.You can find the change in the revised manuscript.

Comments 5: References:

Please check the formatting according to the journal requirements. Sometimes the names of journals are written as abbreviations, and sometimes as full names.

Response 5: Thank you for pointing this out. I agree with your comment. Therefore, I have revised the journal format as you can find the change in the revised manuscript from line 328 to 428.

4. Response to Comments on the Quality of English Language

5. Additional clarifications

Again, we really appreciate your dedicated work on scientific knowledge sharing for the future progress of the research.
Thank you very much.
